# Congenital Haemostasis Disorders and Urology Surgery: Is It Safe?

**DOI:** 10.3390/jcm13082357

**Published:** 2024-04-18

**Authors:** Antoine Bres, Thibaut Waeckel, Yohann Repesse, Xavier Tillou

**Affiliations:** 1Urology Department, CHU de Caen, avenue de la Côte de Nacre, 14000 Caen, France; antoine.bres.kb@gmail.com (A.B.); thibaut.waeckel@icloud.com (T.W.); 2Hematology Laboratory, CHU de Caen, 14033 Caen, France; repesse-y@chu-caen.fr

**Keywords:** urology, haemophilia, surgery, bleeding, complications

## Abstract

Background: There are no specific recommendations for the management of patients with bleeding disorders (BD), such as haemophilia A (HA), haemophilia B (HB), or von Willebrand disease (WD), in urology surgery. **Methods:** We conducted a retrospective study of 32 patients with HA, HB, or WD of any severity. Fifty-seven procedures were performed between January 2017 and September 2023. Surgical interventions were divided into two groups: those with and without electrocoagulation. The control patients were successively matched in a 2:1 ratio. **Results:** The study group consisted of 30 men and 2 women, with 23 HA, 2 HB, and 7 WD. The median age of the patients was 69 years. The BD group had a longer hospital stay of 4 days compared to 1 day (*p* < 0.0001). The incidence of bleeding events was 21% versus 2% (*p* < 0.0001), and the incidence of complications was 21% versus 7% (*p* = 0.0036) for Clavien 1–2 respectively. In the subgroup with intraoperative coagulation, the readmission rate at 30 days was higher (17% vs. 3%, *p* = 0.00386), as was the transfusion rate (17% vs. 3%, *p* = 0.0386). **Conclusions:** This study showed that urological procedures in patients with bleeding disorders were associated with a higher risk of bleeding and complications.

## 1. Introduction

Primary haemophilia and von Willebrand’s disease are the two most common congenital conditions (1 in 5000 people with haemophilia A and 1 in 10,000 people with von Willebrand’s disease) [1,2], causing excessive bleeding. They affect the patient by altering coagulation. Until the 1970s, these patients had a shortened life expectancy [3] due to complications associated with excessive bleeding. Surgery was then considered a last-resort treatment option. Progress in available therapeutics allows for extended life expectancy and safer outcomes after surgery [4]. Therefore, doctors must manage this type of medical case more often and in older patients. It is complex and requires a thorough assessment of the patient’s medical history before surgery [5].

To date, there are no guidelines for the management of patients with bleeding disorders, particularly those undergoing urological surgery. Few studies have reported the results of focused surgical procedures, such as transurethral prostate or bladder resection [6], bladder and prostate biopsies, and prostatectomy [7]. None of the studies included all patients who received care in the urology surgery department, regardless of the surgical procedure. 

This study aimed to compare the surgical outcomes between patients with bleeding disorders in real-life conditions and those without bleeding disorders undergoing urological surgery.

## 2. Materials and Methods

### 2.1. Population

This was a real-life, retrospective, monocentric, matched-pair study. Patients with von Willebrand disease and haemophilia of all severities were included. All surgeries performed in the urology department were analysed. Interventions were divided into two subgroups: interventions with or without an available electrocoagulation device. All patients with haemophilia or von Willebrand’s disease who underwent surgery in our tertiary department of urology between January 2017 and September 2023 were included. Thirty-two patients with bleeding diathesis and matched pairs were analysed for duration of hospital stay (main criteria), administration of coagulation factors, haemoglobin before and after surgery, need for blood transfusion, complications (using the Clavien–Dindo scale), and eventually the need for readmission. Readmission included admission to the ER or hospitalisation in the urology department, which had to occur within the month following surgery. The number of days between surgery and readmission was also recorded. Two subgroups were analysed: surgery allowing coagulation and surgery without direct access to coagulation (prostate biopsy, ureteroscopy, and urethrotomy). TURP (transurethral resection of the prostate) was performed using a “greenlight” laser. These two subgroups were compared using the same criteria as previously described. Non-inclusion criteria were age < 18 years, secondary bleeding disorder (acquired antibodies against factor VIII or IX), and follow-up of less than 60 days after surgery. Matched pairs were selected if they had undergone the same surgery in the same year, with a difference in age of <10 years. The samples were matched in a ratio of two for one patient with a bleeding disorder.

### 2.2. Bleeding Disorders and Treatments

Haemophilia is defined as basal factor VIII (haemophilia A) or factor IX (haemophilia B). It is classified as severe when <1%, moderate when between 1% and 5%, and mild when above 5%. Patients treated between 2017 and 2018 received Advate^®^, a recombinant factor VIII. Patients treated in the following years received Elocta^®^ because it offers a longer duration of action. Von Willebrand’s disease is defined as a defect in the quantity or quality of the von Willebrand factor. Types 1 and 3 are quantitative defects assessed by the level of VWF (type 1 < 50%; type 3 < 1%), whereas type 2 is a quality defect measured by the VWF: RCo ratio (which correlates with the activity of VWF). The targeted coagulation factor level was ≥ 100% before surgery. Desmopressin (1-deamino-8D-arginine vasopressin, dDAVP) is a synthetic analogue of the antidiuretic hormone that releases VWF stored in endothelial cells. It can be administered intravenously (Minirin^®^) or nasally (Octim^®^). After one administration, there is an increase in VWF and factor VIII by a factor of 2 to 5 in 30 to 60 min. The return to the base rate occurs between 6 and 9 h. Our centre is a tertiary care structure. It is the regional centre for bleeding disease (CRTH) in our region and is recognised as a centre for resources and proficiency in bleeding disease (CRC-MHR). All included patients underwent prior evaluation by a member of the haemostasis department, who provided a customised protocol regarding coagulation factor supplementation. The choice of blood factor supplementation was based on previous bleeding events, the type of bleeding diathesis, the basal blood factor level of the patient, and the type of surgery. An initial protocol adapted to the pathology was provided by the haemostasis team (generally for D − 1, D0 and D + 1 and D + 2), then the rest of the general supplementation for D − 1, D0 and D + 1 and D + 2 was conditioned by the clinical and biological results. Patients who experienced bleeding or who were rehospitalised had longer durations of supplementation than those who did not. Each protocol differed depending on the surgery and procedure envisaged. The protocol was tailored for each patient. The targeted factor rate is generally the same, that is, greater than or equal to 100%.

Factor VIII and factor IX activities (FVIII:C and FIX:C) were determined using a 1-stage assay only. The 1-stage assay is an activated partial thromboplastin time (aPTT) assay. We used Synth Asil^®^, Werfen processed on a KC10A coagulation analyser (Trinity Biotech, Lemgo, Germany) with FVIII-deficient plasma from Werfen^®^ until December 2020, then Actin FS from Siemens was used. In all patients with haemophilia, the preoperative factor levels were ≥100%. 

### 2.3. Statistical Analysis

Data was collected using Microsoft Excel (Albuquerque, NM, USA). The quantitative variables are listed in the tables as follows: number of patients (N), median ± standard deviation, first interquartile range, and third interquartile range. Statistical analyses were performed using GraphPad Prism for Windows^®^ (version 9.4.0; GraphPad Software, San Diego, CA, USA). Categorical variables were analysed using the chi-square test and Fisher’s exact test when applicable. Continuous variables were analysed parametrically using the Student’s *t*-test and non-parametrically using the Kruskal–Wallis or Mann–Whitney test.

The results were considered statistically significant at *p* < 0.05. Data collection followed the French legislation concerning prospective interventional studies to evaluate routine care (Article Art. L1121-1-2 of the French Public Health Code (Code de santé publique français)). The present study protocol was reviewed and approved by the Institutional Review Board of our University Hospital (no approval number for retrospective studies at the time when the study was carried out). Informed consent was obtained from all participants when they were enrolled (use of medical data).

## 3. Results

### 3.1. Description of the Study Group

As shown in Table 1, the patients were aged between 18 and 81 years old. The mean age in the study group was 69 years (65; 75), while it was 69.5 (57; 76) in the control group. The sex ratio was 15 in the study group and 15.3 in the control group. 

As seen in Table 2, among patients with haemophilia, three had a severe form. Four patients had a moderate form, and four had a mild form. Among the six patients with von Willebrand’s disease, three had type 1 disease, and the other three had type 2 disease.

### 3.2. Urology Surgeries

Fifty-seven procedures were performed in total. Among the 29 interventions involving electrocoagulation, the most common were laparoscopic prostatectomy (n = 7) and transurethral resection of the bladder (TURB) (Table 3). In the no electrocoagulation subgroup, the most common intervention was prostate biopsy (n = 11).

### 3.3. Factor Suppletion

Table 4 shows how factor supplementation protocols were assigned to patients according to the type and severity of haemophilia and type of surgery. In the no-coagulation subgroup, only one patient did not receive recombinant coagulation factor supplementation. The patient was hospitalised twice for bladder chemotherapy via a bladder catheter. In the coagulation subgroup, one patient received desmopressin before undergoing vasectomy, and one patient had no treatment before orchidopexy. The last patient was not treated because his basal factor VIII activity level was above 70% before surgery. Other patients received coagulation factors (VIII or IX) according to the type and depth of haemophilia and the type of surgery. There is no consensus regarding patient management according to the type of surgery described; therefore, the referring haematologist must formulate an individualised and adapted protocol.

### 3.4. Hospital Stay

As shown in Table 5, the median hospital stay was significantly longer in the study group (4 (3; 6) days) than in the control group (1 (1; 2) day; *p* < 0.0001). The same result was observed in both subgroups (with or without coagulation, *p* < 0.0001). We also observed that the median hospital stay was longer in the coagulation subgroup, although the difference between subgroups was not tested. 

There were few serious complications in the study group. The extension of stay was linked, in the vast majority, to the need for the administration of factors and clinical-biological monitoring requested by the haemostasis team. Out of 39 stays with haemophiliacs/Willebrand, we had 4 short/ambulatory stays and 35 extended stays. Among the 35 extended cases, 13 (37%) were due to complications.

### 3.5. Bleeding Events and Blood Transfusions

The number of bleeding events was significantly higher in the study group (21% vs. 2%, *p* < 0.0001) in both subgroups (24% vs. 2%, *p* < 0.0001 in the electrocoagulation group and 18% vs. 2% in the no electrocoagulation group, *p* < 0.001). The most frequent bleeding event was haematuria in both the study (n = 9, 75%) and control (n = 2, 100%) groups. The number of transfusions was higher in the study group (11% vs. 2%, *p* = 0.0173). This result was not observed in the electrocoagulation subgroup (*p* = 0.33). Transurethral bladder resection was most frequently associated with postoperative transfusion (n = 3, 37.5%). These events were responsible for the longer hospital stays in the study group (37%). We were unable to analyse the difference in pre- and post-surgery haemoglobin levels due to the lack of data in the control group. However, the median decrease in the study group was only 1 g/dL.

### 3.6. Readmission and Complications

The readmission rates were not significantly different between the study groups (7% vs. 4%; *p* = 0.4431). The most frequent reason for readmission was haematuria (n = 5, 62.5%), which was also observed in the study (n = 3, 75%) and control (n = 2, 50%) groups. The second most frequent reason was postoperative infection (n = 2; 25%). There were more postoperative complications in the study group (21% vs. 7% in the electrocoagulation subgroup and 2% vs. 1% in the no electrocoagulation subgroup, *p* = 0.0036).

## 4. Discussion

This study aimed to compare the surgical outcomes between patients with von Willebrand’s disease and haemophilia in real-life conditions and those without bleeding disorders undergoing urological surgery. Our study represented everyday urologist activities, and all types of surgery were represented from endourology to robotic surgery. The three most common surgeries were prostate biopsy, laparoscopic prostatectomy, and TURB. The length of hospital stay (LOS) and number of bleeding events were significantly higher in our study group than in the control group. We divided the surgical procedures into two subgroups: those with and without access to electrocoagulation. This allowed us to focus on the impact of these devices during the surgery. This subgroup analysis showed that there was less need for postoperative transfusion and fewer readmissions in the no electrocoagulation subgroup. A longer LOS for patients with bleeding disorders was also found in a previous matched-pair analysis [6] of TURP (12.08 days vs. 5.83 days) and TURB (11.15 days vs. 6.15). However, the median length of hospital stay was shorter in this study. This is probably due to the use of a green light laser for TURP, which is now recommended for patients undergoing antiplatelet or anticoagulation therapy [8]. This can be extended to patients with bleeding disorders. A similar result was found in a study [9] comparing patients with haemophilia and controls undergoing abdominal surgery. The median hospital stay for laparoscopic prostatectomy was 7 days and 3 days in the study and control groups, respectively. A systematic review [10] of patient outcomes after laparoscopic prostatectomy showed a median LOS of 1.9 days. Another study on haemophilia [11], which compared open surgery and laparoscopic surgery in 2014, showed no significant differences between the study group and matched pairs. However, the median LOS was 6 days for laparoscopic surgery, 8.5 days for open surgery, and 6 days for laparoscopic surgery in the control patients. Regarding the incidence of bleeding events, it would be possible to decrease the length of hospital admission for a prostate biopsy. However, some patients would still benefit from longer surveillance periods after TURB or laparoscopic prostatectomy. In a study comparing patients with haemophilia with a control group undergoing abdominal surgery [9], the rate of postoperative bleeding in the study group was 5.5%. It appears that there were more bleeding events in our study (21%) than those in the literature. However, after excluding TURB, this rate decreased to 14%. In another study [6], the rate of bleeding events after TURP in the haemophilia population was 16.7%. In our study, the incidence of bleeding after TURP was 20%. This number is similar to that reported in other studies [7]. There were four readmissions for gross haematuria (7%) in the study group. The surgical procedure most associated with readmission was TURB, with two out of six readmissions. A study [12] on the outcome of TURB in 14,514 patients without bleeding disorders showed a complication and readmission rate of 4.3%, with 2.1% gross haematuria. In both groups, most complications were classified as minor (Clavien 1–2), and there were significantly more complications in the study group. Surprisingly, two major adverse events in the study group (Clavien 4b and 3b) were observed in the electrocoagulation subgroup. The first was a major retroperitoneal haematoma associated with acute anaemia after laparoscopic prostatectomy with node dissection that required intensive care, blood factor infusion, and blood transfusion. The second reason was profuse gross haematuria after transurethral bladder resection, which had to be managed by performing bladder coagulation associated with blood factor infusion and blood transfusion. Both patients had haemophilia A, the first with a mild form, and the second with a severe form. Bleeding risk scores such as those from the International Society of Thrombosis and Haemostasis and Scientific and Standardization Committee (ISTH/SSC) [13] are available online but are rarely used in practice. Several drugs are available for factor VIII supplementation in the case of haemophilia A. The choice of molecule is driven by pharmacological improvements. Patients treated between 2017 and 2018 received Advate^®^, a recombinant factor VIII. Those treated in the following years received Elocta^®^ because it offers a longer duration of action [14]. Desmopressin has only been used for short-duration interventions in patients with mild haemophilia or type 1 Willebrand disease. This allows for the release of coagulation factors by endothelial cells, offering a short enhancement in coagulation and avoiding the injection of recombinant factors [15]. The major limitation of this study was that it was a single-centre retrospective study with a small sample size. This is due to the low frequency of haemophilia and Willebrand disease [16]. The transmission mechanism of haemophilia [17] and the population in urology are responsible for the higher number of men than women (sex ratio = 15). To our knowledge, this is the first study that includes such a large panel of interventions.

## 5. Conclusions

This study showed that, compared to other patients, urological procedures in patients with bleeding disorders are still associated with a higher risk of bleeding and complications. The incidence of postoperative adverse events varied significantly according to the type of surgery performed. We showed that surgeries with access to electrocoagulation were more frequently associated with transfusion and readmission, particularly TURB and laparoscopic prostatectomy. Most of the complications were low-grade (Clavien 1–2). In some cases, such as prostate biopsy, the LOS can be lowered to a single postoperative night; however, these patients must be carefully selected. Patients with bleeding disorders must be directed towards an expert centre because of the availability of logistic and human resources. Only these structures have teams of trained haematologists, anaesthetists, and urologists, who can work together to provide ideal perioperative management.

## Figures and Tables

**Table 1 jcm-13-02357-t001:** Characteristics of the population.

	Study Group	Control Group	*p*
Number of patients	32	114	NA
Median age (years)	69 (65;75)	69.5 (57;76)	0.7480
(with electrocoagulation)	68 (64.5; 74.5)	69.5 (49.75; 76)	0.9126
(without electrocoagulation)	69.5 (65; 75)	60.25 (69.5; 75.75)	0.5985
Sex			
male	30	107	
female	2	7	
Sex ratio	15	15.3	NA

**Table 2 jcm-13-02357-t002:** Description of the study group.

Patients	N=
Haemophilia A:	23
mild	12
moderate	5
severe	6
Haemophilia B:	2
mild	2
moderate	0
severe	0
Von Willebrand’s Disease:	7
Type 1	4
Type 2	3
Total	32

**Table 3 jcm-13-02357-t003:** Types of surgery.

Types of Surgery	Study (N)	Control (N)
**Surgery with electrocoagulation**		
Laparoscopic prostatectomy	7	14
Transurethral resection of prostate (TURP)	5	10
Transurethral resection of bladder (TURB)	6	12
Urinary artificial sphincter	1	2
Hydrocelectomy	1	2
Removal of epididymal cyst	2	4
Orchidopexy	1	2
Vasectomy	4	8
Circumcision	2	4
Sub total	29	58
**Surgery without electrocoagulation**		
Prostate biopsy	11	22
Ureteroscopy	3	6
Ureteral stent	1	2
PCNL	1	2
Urethrotomy	3	6
Cystoscopy	4	8
Bladder instillation	5	10
Sub-total	28	56
Total	57	114

**Table 4 jcm-13-02357-t004:** Blood factor supplementation.

	Without Electrocoagulation	With Electrocoagulation
Procedures	18	25
**Haemophilia A**	16	23
rFVIII	16	20
Basal FVIII:C	<1–35%	<1–42%
Other	0	3 (desmopressin = 2)
Basal FVIII:C		26–39%
**Haemophilia B**	4	2
rFIX	2	2
Basal FIX:C	17%	17–32%
Other	2 (no protocol)	0
Basal FIX:C	32%	

Other: desmopressin or no protocol at all. Abbreviations: rFVIII, rFIX—recombinant factor VIII or IX; basal FVIII:C, FIX:C—base-level activity of factor VIII or factor IX.

**Table 5 jcm-13-02357-t005:** Group comparison.

	Study Group	Control Group	*p*
Median Hospital stay (days)	4 (3; 6)	1 (1; 2)	<0.0001 (M)
Electrocoag	6 (3; 9.5)	2 (1; 3]	<0.0001 (M)
No Electrocoag	4 (2.25; 4.75)	1 (1; 1)	<0.0001 (M)
Bleeding event	12 (21%)	2 (2%)	<0.0001 (F)
Electrocoag	7 (24%)	1 (2%)	0.0016 (F)
No Electrocoag	5 (18%)	1 (2%)	0.0145 (F)
Transfusion	6 (11%)	2 (2%)	0.0173 (F)
Electrocoag	5 (17%)	2 (3%)	0.0386 (F)
No Electrocoag	1 (3%)	0 (0%)	0.3333 (F)
Re admission (all)	4 (7%)/6 (10%)	4 (4%)	0.4431
Electrocoag	3 (10%)	2 (3%)	0.3280
No Electrocoag	1 (4%)	2 (4%)	>0.9999
Complications (all)			0.0036 (X)
Clavien 1–2	12 (21%)	8 (7%)	
Clavien 3–4	2 (2%)	1 (1%)	
Electrocoag			0.0568 (X)
Clavien 1–2	6 (21%)	5 (9%)	
Clavien 3–4	2 (7%)	1 (2%)	
No Electrocoag			0.0248 (X)
Clavien 1–2	6 (21%)	3 (5%)	
Clavien 3–4	0 (0%)	0 (0%)	

(M) Mann–Whitney test. (F) Fisher’s test. (X) Chi-square test for trends.

## Data Availability

Shared upon reasonable request at xavtillou@gmail.com.

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
