# Peer review of "Congenital Haemostasis Disorders and Urology Surgery: Is It Safe?"

_jcm, 2024, doi:10.3390/jcm13082357_

Round 1

Reviewer 1 Report

Comments and Suggestions for Authors

In this retrospective, single center study, the authors have attempted to describe the bleeding events and complications seen with common urologic procedures in patients with underlying bleeding diathesis.  Overall a very well written article.

Few comments:

1.Would be helpful to know what caused the prolonged jhospital stay in the study group:

-was it for factor repletion or complication? 

- median day after procedure bleeding event was noted.

- Since it was individualized plan was there a difference in duration of factor replacement between bleeders and non bleeders? 

Since it is observational, retrospective study the option may be limited but would be helpful to know what  can be inferred from bleeders and some strategies to mitigate it. 

Also Lines 193-196 Please give reference. 

Also throughout the discussion "," is used instead of decimal. Please correct.

Author Response

Dear reviewers

Thank you for your time reading our manuscript.

Please find below our responses to your comments

Reviewer 1

In this retrospective, single center study, the authors have attempted to describe the bleeding events and complications seen with common urologic procedures in patients with underlying bleeding diathesis.  Overall a very well written article.

Few comments:

1.Would be helpful to know what caused the prolonged hospital stay in the study group:

-was it for factor repletion or complication? 

- median day after procedure bleeding event was noted.

Response: There were few serious complications in the study group. The extension of stay was linked in the vast majority to the need for the administration of factors and clinical-biological monitoring requested by the haemostasis team. Out of 39 stays with hemophiliacs/Willebrand, we had 4 short/ambulatory stays and 35 extended stays. Among the 35 extended cases, 13 were due to complications (37%).

We have added this data to the revised manuscript.

- Since it was individualized plan was there a difference in duration of factor replacement between bleeders and non bleeders? 

Response: Yes, an initial protocol adapted to the pathology was provided by the hemostasis team (generally for D-1, D0 and D+1 and D+2 then the rest of the general supplementation for D-1, D0 and D+1 and D+2 then the continuation of the supplementation was conditioned by the clinical and biological results. Patients who experienced bleeding or who were rehospitalised had longer durations of supplementation than those who did not.

We have added this information to the Materials and Methods section.

 Since it is observational, retrospective study the option may be limited but would be helpful to know what  can be inferred from bleeders and some strategies to mitigate it. 

Response: As a note in the conclusion, it seems to us that particular attention must be paid to TURBT and prostatectomy because these are two surgeries from which we do not necessarily expect the most bleeding and nevertheless, in the study, caused the most postoperative complications. To limit complications, in my opinion, we must agree to keep patients for longer and not consider them as the general population even though they receive supplements (or set up a system like in an outpatient setting with daily calls to get news)

Also Lines 193-196 Please give reference. 

Response: Done. It is reference nb 11

Also throughout the discussion "," is used instead of decimal. Please correct.

Response: Done

Reviewer 2 Report

Comments and Suggestions for Authors

Dear authors, hereafter my suggestions

First of all, English could be improved overall. Some parts of the article should be also reviewed par an hematologist/ specialist in haemostasis (see below)

- lane 33 you stated that no guidelines exist for urological surgery, but extensive guidelines do exit fort patients with bleeding disorders and surgery in general (f.i.see your first references HAS- Urological surgery is classified s surgery with high risk pf bleeding). It should be at least mentioned.

- lane 69-93: the paragraph on bleeding disorders and treatment must be reviewed by an specialist in haemostasis, because many statements are really imprecise (classification of VWD, use of DDAVP: for which patients, laboratory determination of factor levels)

- lane 112 Results

the text does not correspond to  table 2:  figures concerning severity of HA patients

-lane 125 TURB and TURP are not defined and should be mentioned in table 3, for the better comprehension

- lane 130..... 140 to be reviewed by an hematologist. Initial levels of factors are not specified for patients undergoing surgery as well as factor levels to be targeted for specific intervention. Mean amount of factor administrated could be of interest.  This paragraph should be reviewed. f.i To take care of an severe or moderate hemophiliac is really different!

Lane 154-155 the number of bleeding events was significantly higher in the study group, whatever the type of procedure (electrocoag or not), on the contrary this difference was not observed for the number of transfusions

But in lane 181 you state that transfusion were less in the no electocoag group?

Lane 211-216 Font is different to the rest of the text. Were the two problematic patients well supplemented? How was the post-op management for these problematic patients. ISTH score is necessary highly positive in hemophiliac patients (to be managed by a specialist)

Lane 216 to 231 to be reviewed by a specialist; whatever the treatment if the patient is well prepared, it should be theoretically OK, quality of surgery could also be a cause of bleeding

for the discussion: Essential point is the multidisciplinary  interaction of urologist, haematologist, and blood bank staff

Comments on the Quality of English Language

not publishable as such, in my opinion

best regards, AD

Author Response

Dear reviewers

Thank you for your time reading our manuscript.

Please find below our responses to your comments

Reviewer 2

Dear authors, hereafter my suggestions

First of all, English could be improved overall. Some parts of the article should be also reviewed par an hematologist/ specialist in haemostasis (see below)

Response: The manuscript was edited by a native English-speaking medical doctor. One of the authors is a professor of haematology, the head of the department. However, we decided to orient the manuscript for readability by both urologists and haematologists. Therefore we did not provided too deep details

- lane 33 you stated that no guidelines exist for urological surgery, but extensive guidelines do exit fort patients with bleeding disorders and surgery in general (f.i.see your first references HAS- Urological surgery is classified s surgery with high risk pf bleeding). It should be at least mentioned.

Response: We confirm that there were no specific guidelines for urological surgery. Urology is a medical field in which various types of surgery are performed, including endoscopic, percutaneous, transrectal, coelioscopic, and laparotomy. This makes the management of patients with haemophilia challenging, and each patient must receive tailored treatments according to the disease, surgical strategy, and expected complications. The guidelines (reference N°1) provide only general recommendations and factor-level targets for surgery without specifying the type of surgery.

- lane 69-93: the paragraph on bleeding disorders and treatment must be reviewed by an specialist in haemostasis, because many statements are really imprecise (classification of VWD, use of DDAVP: for which patients, laboratory determination of factor levels)

Response: For the technical details we reviewed this part with Prof Repesse for the names of the reagents used in the laboratory, We put what seemed reasonably technical to us. For the classification of Willebrand disease, VWD has many subdivisions which do not seem relevant to this study. (Do you think we should enrich the classification at the risk of cluttering the article?)

- lane 112 Results : the text does not correspond to  table 2:  figures concerning severity of HA patients

Response: Corrected

-lane 125 TURB and TURP are not defined and should be mentioned in table 3, for the better comprehension

Response: Corrected

- lane 130..... 140 to be reviewed by an hematologist. Initial levels of factors are not specified for patients undergoing surgery as well as factor levels to be targeted for specific intervention. Mean amount of factor administrated could be of interest.  This paragraph should be reviewed. f.i To take care of an severe or moderate hemophiliac is really different!

Response: Each protocol differed depending on the surgery and procedure envisaged. The protocol was tailored to each patient. The targeted factor rate is generally the same, that is, greater than or equal to 100%. We spoke about it with Prof. Repesse,  but there are no clear recommendations for supplements. Haematologists do this according to their habits.

Response: We have added these details to the Materials and Methods section.

Lane 154-155 the number of bleeding events was significantly higher in the study group, whatever the type of procedure (electrocoag or not), on the contrary this difference was not observed for the number of transfusions

But in lane 181 you state that transfusion were less in the no electrocoag group?

Response: Yes correct. This means that by subgroup analysis,  having an electrocoagulation device during surgery lowered the need for transfusions.

Lane 211-216 Font is different to the rest of the text. Were the two problematic patients well supplemented? How was the post-op management for these problematic patients. ISTH score is necessary highly positive in hemophiliac patients (to be managed by a specialist)

Response: In the limited number of words available, we cannot go into all the details of the support. Regarding complications after RP, the patient was discharged too early, which caused problems in a patient who presented with a very large retro peritoneal haematoma.

Lane 216 to 231 to be reviewed by a specialist; whatever the treatment if the patient is well prepared, it should be theoretically OK, quality of surgery could also be a cause of bleeding

for the discussion: Essential point is the multidisciplinary  interaction of urologist, haematologist, and blood bank staff

Round 2

Reviewer 2 Report

Comments and Suggestions for Authors

manuscript is well improved

some minor remarks remains

table 1

I don't understand the meaning of the lanes with and without are they useful?

Lane 127 to 129 figures in the table and int he text are still bot corrected, they don't correspond to each other

lane 228 to 233 FONT of this paragraph is different from the rest of the manuscript

Lane 243 what do you mean by exogenous factors? plasmatic factors? other . It is not clear to me

Author Response

Dear reviewer

Please find below our responses to your comments

table 1

I don't understand the meaning of the lanes with and without are they useful?

Answer: This means with or without electrocoagulation during surgery. We added it the table.

Lane 127 to 129 figures in the table and int he text are still bot corrected, they don't correspond to each other

Corrected

lane 228 to 233 FONT of this paragraph is different from the rest of the manuscript

Corrected

Lane 243 what do you mean by exogenous factors? plasmatic factors? other . It is not clear to me

Answer: We have removed the sentence and rewritten it.